# OpenReview forum: "Safety-Tuned LLaMAs: Lessons From Improving the Safety of Large Language Models that Follow Instructions"
_ICLR.cc/2024/Conference — ICLR 2024 poster_

### Official Review · Reviewer_gf5T · 2023-10-28

**Soundness:** 3 good
**Presentation:** 3 good
**Contribution:** 2 fair
**Rating:** 6
**Confidence:** 4

**Summary:**

The authors created a dataset for evaluating improper behavior of LLMs and the trade-off between safety and helpfulness objectives.
They report on experiments with different fine-tuning variants of LLAMAs, showing some unsurprising results (there is a trade-off between safe behavior and helpful behavior, the amount of labeled data matters) and some less obvious results (asking the model to provide an opinion yields safer answers and asking to answer a question to provide instructions; finding the right balance between safety and helpfulness is difficult).

**Strengths:**

Experiments seem sound and done carefully. Not all results are trivial and are thus worthy of being reported in a paper. The dataset could be useful for further evaluation of alignment and safety methodologies.

**Weaknesses:**

The claims on page 2 need to be redone. The ones that are shown are (a) not novel enough (e.g. there is a tension between the two objectives) and (b) redundant (2 and 3 are different ways of stating 1).

However, there are real contributions (such as the creation of the dataset and the observed experimental results) which should be highlighted in the claims.

Although the experiments may help to adjust current LLMs in a slightly safer way, this may all be useless if the overall LLM with RLHF or safety fine-tuning is very easy to break with a little bit of appropriately chosen fine-tuning, e.g. see Qi et al "Fine-tuning Aligned Language Models Compromises Safety, Even When Users Do Not Intend To!", arXiv:2310.03693

**Questions:**

How do the authors put their work in the perspective of the results from Qi et al 2023 (arXiv:2310.03693)?
Is it even worthwhile pursuing this kind of avenue if fine-tuning for safety can be completely broken so easily?

---

> ### Author Response · Authors · 2023-11-17
>
> We thank the reviewer for the comments and the review.
>
> **The claims on page 2 need to be redone.**
>
> We thank the reviewer for the comment, we have modified the section to make our claims clearer, but we still believe that the general contributions remain the same:
>
> Claim 1 is about the tension and our paper is the first to show this in an entirely reproducible setting in which both models and data are open-sourced.
>
> Claim 2 is connected to the fact that we believe that safer models can be obtained with little effort (5% of the data), suggesting that most models can be made safer. This happens without completely compromising general capabilities; this would suggest that thanks to safety tuning there is little to no tension (claim 1), however…
>
> Claim 3 is about the fact that safety training can introduce a novel phenomenon, called exaggerated safety. This is different from the other claims, as it shows that even if general capabilities do not seem to decrease (claim 2) there is still an impact that comes from safety training.
>
> **"Although the experiments may help to adjust current LLMs in a slightly safer way, this may all be useless if the overall LLM with RLHF or safety fine-tuning is very easy to break with a little bit of appropriately chosen fine-tuning, e.g. see Qi et al "Fine-tuning Aligned Language Models Compromises Safety, Even When Users Do Not Intend To!", arXiv:2310.03693"**
>
> Thanks for pointing out the work by Qi et al. In their work, they show that fine-tuning can be used to break safety capabilities, which we understand is an important finding.
>
> However, we believe it is fundamental to work on releasing safer models. With the current models, it is very easy for non-expert users to create toxic and unsafe content. Fine-tuning still requires a reasonable amount of expertise most users do not have. Our paper focuses on limiting existing problems.
>
> In general, we do not believe that adding safety to the models is a futile experiment, even when it can be broken. Ease of access to unsafe behaviors matters. Safety by default still makes a difference for user-facing applications, even if safety measures can be easily circumvented
>
> We have found that most models in online repositories that are instruction-tuned lack any safety guarantee. This suggests that, as of today, there is no need to “break the safety” as the models released are already unsafe. In our paper, we propose that safety tuning can be successfully applied to most training models without compromising their general performance, with some caveats we have described in the paper.

---

### Official Review · Reviewer_yQ2D · 2023-10-30

**Soundness:** 3 good
**Presentation:** 3 good
**Contribution:** 2 fair
**Rating:** 6
**Confidence:** 3

**Summary:**

This paper discusses the safety concerns of large language models (LLMs) that focus on helpfulness rather than harmlessness in their instruction-tuning. It reveals that popular instruction-tuned models are highly unsafe, and adding just 3% safety examples can significantly improve their safety. While safety-tuning does not significantly reduce models' capabilities or helpfulness, it can lead to exaggerated safety behaviors, where models refuse perfectly safe prompts if they resemble unsafe ones. The results highlight the trade-offs between training LLMs to be helpful and harmless.

**Strengths:**

+ The paper addresses an important and timely direction, which is the safety concern of LLM
+ The paper is easy to follow in general
+ Figure 1 provides a good overview of the main focus of the paper

**Weaknesses:**

- The collected safety dataset is too small, many are about 100.
- The difference with related work is not deeply discussed, e.g., the reasons for not using the red teaming dataset in safety training are not convincing enough
- There are some parts of the experiment results requiring further explanation

**Questions:**

In general, the idea of the work is interesting, to evaluate the safety of the existing open-source LLMs and find that a small amount of safety demonstrations could improve the safety of the LLM. The manuscript is easy to follow and the idea is not hard to understand. However, I think the biggest problem the paper has is the evaluation is kind of weak, e.g., the size of the dataset is small (many are about 100) and some experiment results are not well-explained. I list the questions below.

1. Section 2: Maybe the authors could define what safety is before a deeper discussion. Safety is a very general concept and without a clear definition, it is hard for the readers to understand what the paper discusses.

2. Page 4:
> We randomly selected 2,000 questions from this dataset and used GPT-3.5-turbo to generate “safe” responses to them.

Why do you choose ChatGPT is generate safe responses and how do you guarantee that? Also, the safety seems only limited to the area that you choose, can they be generalized?

3. Section 3.3: Why not use the red teaming dataset?

4. Figure 3(a): Why safety training is not effective in O-Controversial dataset?

5. Page 8:
> Models’ “opinions” do not reflect models’ behavior.

I don't quite get the meaning of this part.

6. Page 8:
> Too much safety can be detrimental.

Maybe you could go over the dataset and get some more stronger numbers. With example only is not convincing enough.

---

> ### Author Response · Authors · 2023-11-17
>
> We thank the reviewer for the comments and the review.
>
> **Definition of Safety**
>
> Thanks for pointing this out, We have added this to the paper.
> In our context, safety is defined as stopping models from following malicious instructions (I-MaliciousInstructions) and generating toxic content (I-CoNa, I-Controversial). We also test how this safety training generalizes on physical safety instructions.
>
> **O-Controversial Results and Models’ Opinions vs Behaviour**
>
> The O-Controversial dataset contains prompts like “Do you think I should kill someone?”.
> There is still a minor reduction in harmfulness even in O-Controverisal. Looking at the examples, in O-Controverisal, the 2000-safety model explains in detail why it does not believe something has to be done (e.g., killing someone). Instead, in I-CoNA the model responds with direct refusal by the model (“As an AI assistant…”) more times; the reward model seems to give a slightly higher score (more harmful) for the former and a lower for the latter.
>
> Thanks for pointing out the limitations of this section. We have clarified this in the paper. What we want to show is that asking the model “Do you think I should kill myself” is very different from asking the model “Tell me ways how I can kill myself”. This shows an additional weak spot of the helpfulness and harmlessness tension and makes. In addition to this, recent research has been devoted to interpreting and identifying the political leanings of different models: these results suggest that the model might say it does not believe killing someone is a good idea, but would still provide ways to do it.
>
> **"The collected safety dataset is too small, many are about 100."**
>
> The training dataset contains 2,000 safety examples for training. The evaluation datasets are smaller but also more targeted. We created smaller evaluation datasets that we could manually control and check. When considered together, the evaluation datasets count more than 500 examples and cover different safety aspects.
>
> While our datasets cannot be used to evaluate all safety limitations in large language models, we still see that the models fail in many cases. Considering also that the rate of failure changes with the scale of safety tuning, our evaluation datasets have relevant diagnostic power.
>
> “Why do you choose ChatGPT is generate safe responses and how do you guarantee that? Also, the safety seems only limited to the area that you choose, can they be generalized?”
>
> ChatGPT responses are very safe, we manually went through the examples to ensure they were all acceptable for our task. Most of the responses are direct refusal (e.g., “As an AI assistant…”), and some offer more explanations (e.g., “Killing someone is bad because…”)
>
> Regarding the training dataset, we start from a series of RedTeaming attempts that cover a wide variety of possible malicious attempts. These attempts cover a wide variety of issues, like doxxing, privacy violation, bias, and content toxic generation. We believe that our sample is representative of these attempts. Results on our PysicalSafety datasets provide insights into how the models generalize on examples that are out of domain.

---

> ### Author Response · Authors · 2023-11-17
> **(second part)**
>
> **Difference with related work**
>
> Thanks for pointing this out, we have extended the paper to discuss this issue more in detail; we believe that no safety dataset can be used to train instruction models.
>
> There are two main datasets for safety-related tasks: the HH-RLHF (Bai et al., 2022) and the RedTeaming (Ganguli et al., 2022) datasets. However, they come with the following limitations:
>
> * they are not made for tuning; Anthropic’s guidelines advise not to use these datasets for tuning: “Training dialogue agents on these data is likely to lead to harmful models, and this should be avoided.”
> * they are less helpful to induce safety behaviors as they were generated using not safety-tuned models.
>
> We agree with the reviewer about the benefit of better explaining why we created a new dataset. We have added a more in-depth discussion on why we did not use the HH-RLHF or the RedTeaming datasets to the appendix.
>
> In particular, the RedTeaming dataset was not meant to train models as the responses come from models that were trained without safety; there is often no refusal and the dataset contains a wide variety of safe and unsafe replies. The reviewer can also look at some of the examples in the dataset to see that they do not offer any guarantee for safety.
>
> The HH-RLHF dataset should be used only to train reward models, but in some cases it has been used to train dialogue agents. In general, we do not believe the training samples are effective to study safety training. Looking at some of the examples in the dataset (like the one mentioned in the paper) demonstrates this and we believe that this issue affects trained models.  In particular, in the Appendix, we show that while models (e.g., MPT 7B chat) trained on the dataset often provide safe completions, they do not learn to be completely safe or to respond with refusal to unsafe instructions. In our opinion, the main reason why they do not completely refuse some of the malicious instructions is that many examples in the dataset do not have a clear refusal, justifying the need for an additional dataset to study how models learn safety.
>
> All in all, no other safety dataset can be used to train models to be safe. Our dataset is a novel contribution and is the first dataset for safety instruction tuning.
>
> **“Maybe you could go over the dataset and get some more stronger numbers. With example only is not convincing enough.”**
>
> Our results show that training with 2000 additional safety examples makes the models exaggerate the safety of 50% of the test instruction. In the Appendix, we report that there is a linear increase with the additional safety examples we add and we also report a direct comparison between helpfulness, harmfulness, and exaggerated safety of different models.

---

### Official Review · Reviewer_cMF9 · 2023-11-01

**Soundness:** 3 good
**Presentation:** 3 good
**Contribution:** 2 fair
**Rating:** 6
**Confidence:** 4

**Summary:**

This paper investigates approaches for improving safety in large language models that have been fine-tuned to follow instructions. The authors show that popular LLMs can generate unsafe or harmful responses when given malicious prompts. They demonstrate that incorporating even small amounts of safety data during instruction tuning, such as a few hundred examples, can substantially reduce unsafe behaviors without negatively impacting performance on standard benchmarks. However, too much safety data leads to models refusing safe prompts, a problem they term exaggerated safety. The paper releases datasets and tools for evaluating safety issues in language models.

**Strengths:**

- Clearly frames the tension between helpfulness and harmlessness in instruction-tuned LLMs. Shows examples of popular models complying with unsafe instructions.
- Examines the tradeoff between safety and capabilities, finding that even not too much safety data leads to exaggerated safety behaviors where models overly refuse safe prompts.

**Weaknesses:**

- The tradeoff of safety and helpfulness is already a well-known fact.
- The safety training data is limited in scope and size. Scaling up safety data could lead to different conclusions.
- The current models still seem susceptible to simple adversarial attacks or instructions framed differently.
- The paper mainly echo with existing finding in literature without proposing novel methods to solve the safety helpfulness tradeoff.

**Questions:**

Have you continued experiments with scaling up the amount of safety data? If so, did you find any "sweet spots" where safety improves without exaggerated safety effects?
How resilient are the safety-tuned models to variations in the phrasing of unsafe prompts or simple adversarial attacks?

---

> ### Author Response · Authors · 2023-11-17
>
> We thank the reviewer for the review and the comments.
>
> **“The tradeoff of safety and helpfulness is already a well-known fact.”**
>
> We want to highlight that this paper is the first study using open-source datasets to evaluate the effectiveness of safety training in large language models, and how this effectiveness varies with the amount of safety data.
>
> Very little is known about the safety training of both Llama2 and ChatGPT, which are considered to be safer language models. Our study represents the first attempt at replicating safety tuning under conditions where all resources have been made openly accessible. In addition, we have released a novel dataset and demonstrated the scaling behavior of safety training, and highlighted a risk of exaggerated safety.
>
> **Scaling Safety and Risk of Adversarial Attacks**
>
> Thank you for these questions. Our experiments show that even little safety data can result in exaggerated safety effects. Given that we already see exaggerated safety when using some safety tuning data, further scaling up safety tuning would likely just increase these problems.
>
> We acknowledge that the resulting models are not completely safe, but they are significantly safer for normal use. Our experiments with various instructions indicate that the model can be generalized effectively. However, it is important to note that there is still a possibility of the model being affected by adversarial and jailbreaking prompts. In this context, it is worth emphasizing that our goal is to establish an initial layer of safety. The ease of accessing unsafe behaviors is an important consideration. Even though safety measures can be bypassed, having safety as the default setting remains crucial for user-facing applications.
>
> **“The paper mainly echo with existing finding”**
>
> We respectfully disagree with this statement. There is currently no similar finding in the literature for which the datasets are also open-sourced.
>
> This is the first paper to systematically quantify the effect of safety training on exaggerated safety and also the first to show that little safety data is enough to ensure a consistent reduction in the effectiveness of malicious attacks. We also released the open source benchmark and code to support users who want to evaluate safety in their models, which is a valuable contribution.
>
> Thank you very much for your time and review! We would really appreciate it if you could let us know if you have additional questions, and we are happy to further clarify.

---

> > ### Author Response · Authors · 2023-11-22
> > **Thank you for your review! Follow up**
> >
> > Dear reviewer cMF9
> > Thank you again for your time! We just want to see if our response has clarified your questions. We hope you would consider increasing your score if we have answered your questions. Please let us know if you have additional comments and we are happy to follow up. Thanks!

---

### Official Review · Reviewer_HxPh · 2023-11-01

**Soundness:** 3 good
**Presentation:** 3 good
**Contribution:** 2 fair
**Rating:** 6
**Confidence:** 4

**Summary:**

This paper studies the safety-tuning of LLMs. It creates a public fine-tuning dataset for safety-tuning, which is currently lacking. This dataset it presents a study on how mixing safety data and common instruction-tuning data would impact the safety and utility of resulting models. It is shown that, by mixing a small portion of safety data, the models would be much safer while do not suffer from too much utility drops.

**Strengths:**

1. The evaluation is comprehensive, covering a diverse set of safety evaluation datasets with focuses on different aspects of model safety. It also considers multiple approaches to evaluate the quality of the response, evaluating the utility of the models. Human studies are also involved.

2. Contribution of a publicly available safety tuning dataset.

3. The analysis is insightful, providing a lot of practical insights gained from safety-tuning experiments.

**Weaknesses:**

The novelty is my primary concern. It is known that aligned models like ChatGPT and Llama-2 involve instruction tuning with safety data. That's exactly what makes these models safer than unaligned ones. What the paper does is just repeat an experimental study on instruction tuning. This makes the contributions of the paper seemingly insignificant.

However, the details of the safety tuning of ChatGPT/Llama-2 are not publicly transparent. Especially, the safety data are not publicly available, and we neither know how different design choices will impact the effectiveness. From this perspective, this work can be publicly accessible material for the public to understand and reimplement the practice.

**Questions:**

Can you explain more on the novelty of this work given my above concern?

---

> ### Author Response · Authors · 2023-11-17
>
> We thank the reviewer for their review and comments.
>
> We will answer the points raised individually.
>
> **Can you explain more on the novelty of this work given my above concern?**
>
> As mentioned by the reviewer, there is limited information regarding the safety training of Llama2 and ChatGPT, which are considered to be safer language models. Most language models available today are either non safety-tuned or use data from ChatGPT, which allows for inheriting its safety guardrails.
>
> Our study represents the first attempt at replicating safety tuning under conditions where all resources have been made openly accessible.
>
> We released a safety dataset to encourage future research, but our contribution goes beyond that. Previous works have not systematically demonstrated that adding a small number of safety examples to fine-tune a model is enough to provide reasonable safety behaviors. Moreover, this is the first paper to directly quantify the safety vs exaggerated safety tradeoff when training models to follow instructions and be safe at the same time.
>
> In addition to this, we want to point out that all our contributions are released as open-source. This final point will encourage future open model releases to account for safety considerations.

---

> > ### Comment · Reviewer_HxPh · 2023-11-19
> > **Reply to the authors**
> >
> > Thanks to the authors for answering my questions. I generally agree with what the authors said above, but with a few additional comments:
> >
> > > Previous works have not systematically demonstrated that adding a small number of safety examples to fine-tune a model is enough to provide reasonable safety behaviors.
> >
> > In the original paper of Llama 2 [1], they also mentioned that "quality is all you need" for alignment. That said, they showed that a moderate number of high-quality data is sufficient.  In the LIMA paper [2], similar information has been conveyed. It would be better if the authors could tone down a little bit on this claim and make meaningful connections to these related work in the paper.
> >
> >
> > > This is the first paper to directly quantify the safety vs exaggerated safety tradeoff when training models to follow instructions and be safe at the same time.
> >
> > Though the concept of "exaggerated safety" is relatively new, the trade-off between helpfulness and harmlessness is not new and has been broadly discovered. Similarly, it would be good for the authors to tone down a little bit.
> >
> >
> > In general, I agree that this paper makes meaningful contributions, as I summarized in my initial review and also supplemented by the authors. It would be great if the authors could consider my suggestions above.
> >
> >
> >
> > [1] Touvron, Hugo, et al. "Llama 2: Open foundation and fine-tuned chat models." arXiv preprint arXiv:2307.09288 (2023).
> >
> > [2] Zhou, Chunting, et al. "Lima: Less is more for alignment." arXiv preprint arXiv:2305.11206 (2023).

---

> > > ### Author Response · Authors · 2023-11-20
> > >
> > > We thank the reviewer for the additional feedback. We really appreciate the effort that was put into interacting again with us and with the paper.
> > > Following your suggestion, we have toned down the contributions and made better references to previous work. In particular, in the “contributions” section:
> > >
> > > * We now added more discussions and acknowledgment of the previous literature on the topic of helpfulness and harmfulness, which are very important contributions.
> > >
> > > * We now discuss “exaggerated safety” as a component of the helpfulness-harmfulness tradeoff, making it part of a bigger process and less of a standalone section.
> > >
> > > Thank you again for these helpful suggestions, which have really helped us to improve the paper!

---

### Meta-Review · Area_Chair_7UZF · 2023-12-11

**Metareview:**

Understanding the statistical tradeoffs between instruction-tuned capabilities and safety properties is important for LLM alignment research. This paper provides complimentary evidence that instruction tuning compromises safety, that a small amount of safety data goes a long way, but also that the models are prone to becoming overly conservative. The paper's contributions are mostly empirical but include dataset consolidation. As such novelty and experimentation on small datasets were identified as weaknesses.

**Justification For Why Not Higher Score:**

The paper documents useful observations on safety tensions with instruction tuned models, but does not propose a systematic approach to resolve this tension, which would have boosted its novelty. However, safety is hard and the paper can serve as a valuable reference for follow-on work.

**Justification For Why Not Lower Score:**

The paper is not borderline in the sense that 4 reviews scored it above the threshold, with safety insights overriding concerns about novelty.

---

### Decision · Program_Chairs · 2024-01-16

Accept (poster)